# Characteristics of two tsunamis generated by successive $M_w$ 7.4 and $M_w$ 8.1 earthquakes in Kermadec Islands on March 4, 2021

Yuchen Wang[1,2], Mohammad Heidarzadeh[3], Kenji Satake[1], Gui Hu[4]

[1]Earthquake Research Institute, The University of Tokyo, Tokyo, 113-0032, Japan
[2]Japan Agency for Marine–Earth Science and Technology, Kanazawa, Yokohama, 236-0001, Japan
[3]Department of Civil and Environmental Engineering, Brunel University London, Uxbridge, UB8 3PH, UK
[4]Guangdong Provincial Key Laboratory of Geodynamics and Geohazards, School of Earth Sciences and Engineering, Sun Yat-Sen University, Guangzhou, 510275, China

*Correspondence to*: Yuchen Wang (ywang@jamstec.go.jp)

**Abstract.** On March 4, 2021, two tsunamigenic earthquakes ($M_w$ 7.4 and $M_w$ 8.1) occurred successively within 2 h in Kermadec Islands, offshore New Zealand. We examined sea level records at tide gauges located at ~100 km to ~ 2,000 km from the epicenters, conducted Fourier and Wavelet analyses as well as numerical modelling of both tsunamis. Fourier analyses indicated that the energy of the first tsunami is mainly distributed over the period range of 5–17 min, whereas it is 8–32 min for the second tsunami. Wavelet plots showed that the oscillations of the first tsunami continued even after the arrival of the second tsunami. As the epicenters of two earthquakes are close to each other (~ 55 km), we reconstructed the source spectrum of the second tsunami by using the first tsunami as the empirical Green's function. The main spectral peaks are 25.6 min, 16.0 min, and 9.8 min. The results are similar to those calculated using tsunami-to-background ratio method and are also consistent with the source models.

## 1 Introduction

The Kermadec Islands are an island arc in the southwestern Pacific Ocean, formed at the convergent boundary where the Pacific Plate subducts under the Indo-Australian Plate (Figure 1) (Billen et al., 2003). The Kermadec Trench, which accommodates westward subduction of the Pacific Plate beneath the active Kermadec volcanic arc, is identified as the key region for concern regarding seismic and tsunami hazards in New Zealand (e.g., Power et al., 2012). The historical records of earthquakes in Kermadec Islands are short but there are three great earthquakes over the 20th century: the 1917 ($M_w$ 8.6; Morgan, 1918), the 1976 ($M_w$ 8.0; Wyss et al., 1984), and the 1986 earthquakes ($M_w$ 7.9; Lundgren et al., 1989).

On March 4, 2021, two earthquakes occurred successively in Kermadec Islands. The first event ($M_w$ 7.4; foreshock) occurred at 17:41:23 (UTC) whose epicenter was at 29.677° S, 177.840° W, south of Raoul Island in the Kermadec Islands region with a depth of 43.0 km (United States Geological Survey (USGS); https://earthquake.usgs.gov/earthquakes/eventpage/us7000dfk3/executive) (Figure 1). The earthquake generated a small tsunami. The second earthquake ($M_w$ 8.1; mainshock) occurred at 19:28:33 (UTC), approximately two hours after the

foreshock. The epicenter was located at 29.723 º S, 177.279 º W with a depth of 28.9 km (USGS). The epicenters of these two successive tsunamigenic earthquakes are very close to each other (~55 km; Figure 1) and their focal mechanisms are similar; both are thrust earthquakes. The National Emergency Management Agency, New Zealand, issued a tsunami warning for coastal areas of the North Island after the $M_w$ 8.1 earthquake. The tsunami propagated across the Pacific Ocean and reached South America. No casualties were reported. The situation of these two consequent tsunamigenic earthquakes resembles the earthquake events also in Kermadec Islands on January 14, 1976, where two great earthquakes ($M_w$ 7.8 and $M_w$ 8.0) occurred approximately within one hour (Power et al., 2012). The mainshock of the 1976 events ($M_w$ 8.0) generated a moderate tsunami recorded by tide gauges.

The occurrences of two successive earthquakes provide us with a rare opportunity to study their source characteristics by different methods. The spectra of tsunami waveforms at tide gauges contain the effects of source, propagation path, and local topography (Rabinovich, 1997; Heidarzadeh et al., 2016; Cortés et al., 2017). For a single event, it is a common practice to reconstruct tsunami source spectrum from tide gauge records by calculating the ratio of tsunami spectra to background spectra (Rabinovich, 1997). This is based on the assumption that the effects of transmission path on the characteristics of tsunami spectra is small compared to local topographic effects when the tsunami source is not too far away from the observation station (Miller, 1972; Rabinovich, 1997). This assumption can be readily made for two successive tsunami events in Kermadec Islands. If the sources of two earthquakes are close to each other, we can adopt the method of empirical Green's function (EGF) to reconstruct tsunami source spectrum (Heidarzadeh et al., 2016). The smaller tsunami event is adopted as the EGF for the larger one. The spectral deconvolution separates the effects of propagation path and local topography around the tide gauge and gives the source spectrum of the larger tsunami. Heidarzadeh et al. (2016) successfully applied the EGF method to the 2015 ($M_w$ 7.0) and 2013 ($M_w$ 8.0) earthquakes in the Solomon Islands.

Here, we studied the characteristics of tsunamis generated by the two successive Kermadec Islands earthquakes and calculated the source spectrum of the tsunami generated by the mainshock (hereafter, the second tsunami). Fourier analysis was applied to the sea level records at 15 tide gauges. Wavelet analysis was adopted to examine the temporal changes of the dominant spectral peaks. Finally, we used two different methods, tsunami-to-background spectral ratio method and EGF method, to reconstruct tsunami source spectra. We compared two alternative methods of determining tsunami source spectra and also compared their results with USGS source models. This is a unique and rare incident and thus the data and analyses would greatly help to further understand tsunami generation and propagation.

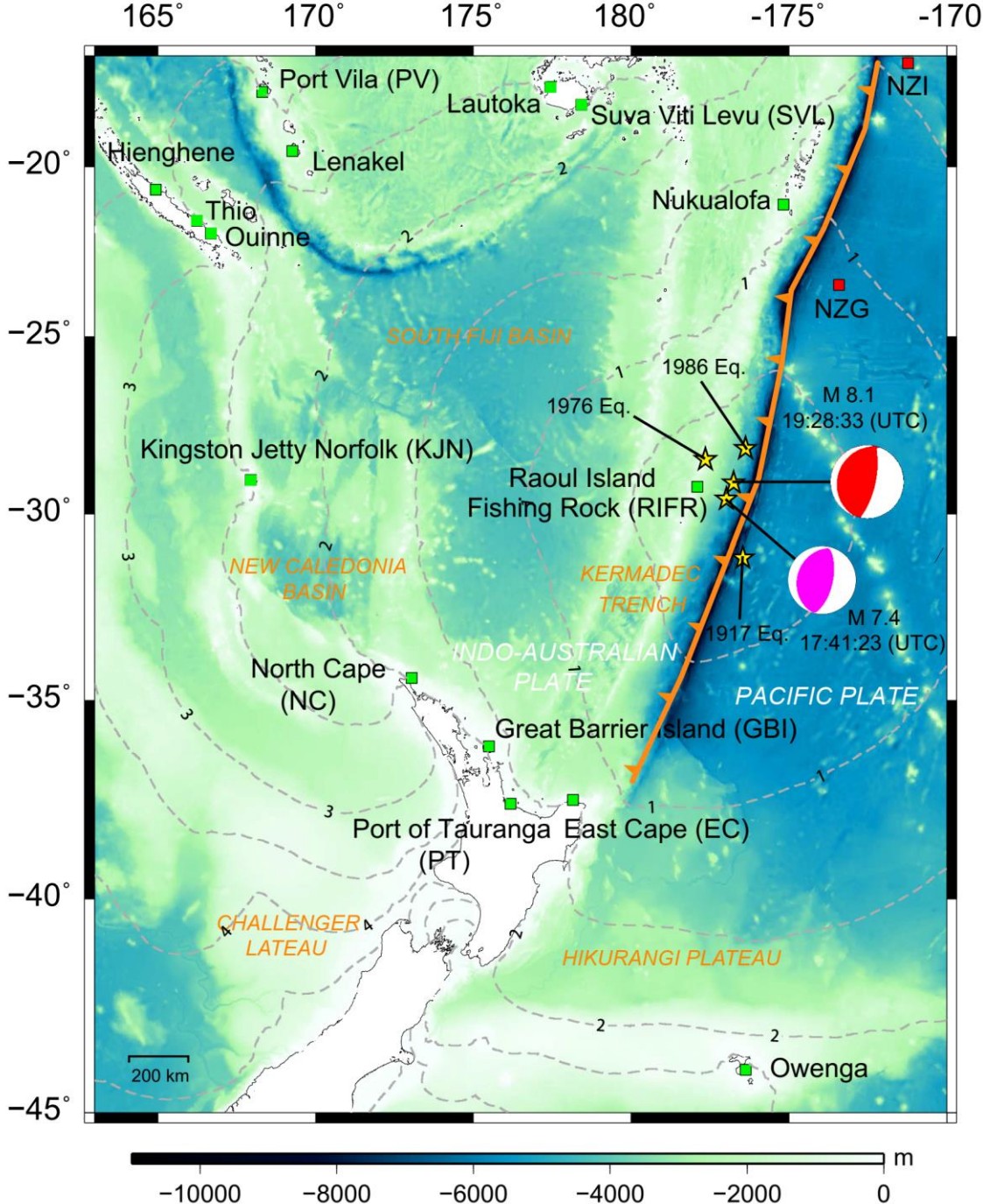

**Figure 1: Bathymetry map of the southwestern Pacific region and the epicenters of two successive earthquakes in Kermadec Islands on March 4, 2021. Pink and red beachballs represent the focal mechanisms (according to the Global Centroid Moment Tensor Project; https://www.globalcmt.org/CMTsearch.html) of the $M_w$ 7.4 and $M_w$ 8.1 earthquakes, respectively. Green squares indicate tide gauges. Red squares indicate Deep-ocean Assessment and Reporting of Tsunamis (DARTs) tsunameters. The travel time of the second tsunami is marked by grey dashed contours with an interval of 0.5 h.**

60

## 2 Data and Method

### 2.1 Tsunami Data

We collected sea level records of 15 tide gauges from the Intergovernmental Oceanographic Commission (http://www.ioc-sealevelmonitoring.org/list.php). These stations are located at distances between ~100 km and ~ 2,000 km from the epicenters (Figure 1). The sampling rate is 60 s. First, we conducted quality control of these data, removing spikes and filling short gaps by linear interpolation. Then, we applied a 2-hour (120-min) high-pass filter to remove the tidal components (Figure 2) (Heidarzadeh and Satake, 2013). Heidarzadeh et al. (2015) showed that the high-pass filtering yields similar results as subtracting calculated tides from the original records. The time series between 17:00:00 (UTC), March 4 and 05:00:00 (UTC), March 5 was selected for analysis. However, as the Raoul Island Fishing Rock (RIFR) tide gauge (Figure 1) lost its data communication during the mainshock, we only used its records before 17:41:00 (UTC) (Figure 2). We note that the data from Deep-ocean Assessment and Reporting of Tsunamis (DARTs) tsunameters of these events were published by Romano et al. (2021). Because the duration of high sampling mode (5 s) is not long enough for spectral analyses at most stations and the data is affected by strong ground motion, we only used the data of NZG and NZI stations (Figures 1 and S1) as a reference to confirm the findings obtained with tide gauges.

### 2.2 Spectral Analyses

Two types of spectral analyses were conducted to tide gauge records: Fourier analysis and wavelet (frequency-time) analysis. For Fourier analysis, a Hanning window with 40% overlaps was applied following the method of Heidarzadeh and Mulia (2021). The tsunami generated by the foreshock (hereafter, the first tsunami) was clearly recorded at eight tide gauges: North Cape (NC), Great Barrier Island (GBI), East Cape (EC), Suva Viti Levu (SVL), Lenakel, Quinne, Kingston Jetty Norfolk (KJN), and Raoul Island Fishing Rock (RIFR), but it is difficult to accurately distinguish the arrival times at these stations. We used the 2 h segment before the arrival of the second tsunami as the input for spectral analyses. For RIFR, we utilized the 2 h segment before the time it lost connection. The second tsunami generated by the mainshock was clearly recorded at all the tide gauges except for RIFR. We selected a 2 h time segment after the tsunami arrival as the input for Fourier analysis. Additionally, background spectra were calculated using the 2 h records before the arrival time of the first tsunami at the same stations. We ensured that there were no storms or other atmospheric events at the time period of the background signals, so the background spectra could exclusively reflect the frequency response of local topography (Cortés et al., 2017; Aránguiz et al., 2019). Tidal components were removed by applying a high-pass filter in a similar way to preparation of the tsunami records (Heidarzadeh and Satake, 2013).

Wavelet analysis was adopted to study the temporal changes of the dominant spectral peaks and the superposition of two tsunamis. We applied the wavelet package provided by Torrence and Compo (1998), which uses the Morlet wavelet

mother function. We only conducted wavelet analysis to seven tide gauges that clearly recorded both tsunamis. The
waveform segments for wavelet analyses are the same as those used for Fourier analyses.

## 2.3 Earthquake Slip Models and Tsunami Numerical Simulation

We used numerical simulation to study the propagation paths of the two tsunamis. For the initial condition, the finite
fault models provided by the USGS were adopted (Slip model of the first event at:
https://earthquake.usgs.gov/earthquakes/eventpage/us7000dfk3/finite-fault; Slip model of the second event at:
https://earthquake.usgs.gov/earthquakes/eventpage/us7000dflf/finite-fault). The source model of the first ($M_w$ 7.4)
earthquake has a rectangular dimension of 80 km (length) × 80 km (width). The strike and dip angles are 196° and 32°,
respectively. The spatial intervals of sub-faults are 4 km × 4 km. The source model of the second ($M_w$ 8.1) earthquake has a
rectangular dimension of 240 km (length) × 190 km (width) with strike and dip angles of 210° and 16°, respectively. The
spatial intervals of sub-faults are 10 km × 10 km. We used Okada's dislocation model (Okada, 1985) to compute the
seafloor deformation and used it as the initial condition for tsunami propagation modeling. The bathymetric grids were
obtained from the General Bathymetric Chart of the Oceans and resampled to 0.9 arc-min. The *JAGURS* numerical package
(Baba et al., 2015) was adopted to simulate the tsunami propagation based on linear long-wave equations. The time step for
numerical simulations is 1.0 s. We simulated two tsunamis separately up to 12 h after the earthquake origin time. Tsunami
travel time (TTT) calculation was performed for the second tsunami by the TTT software of Geoware (2011).

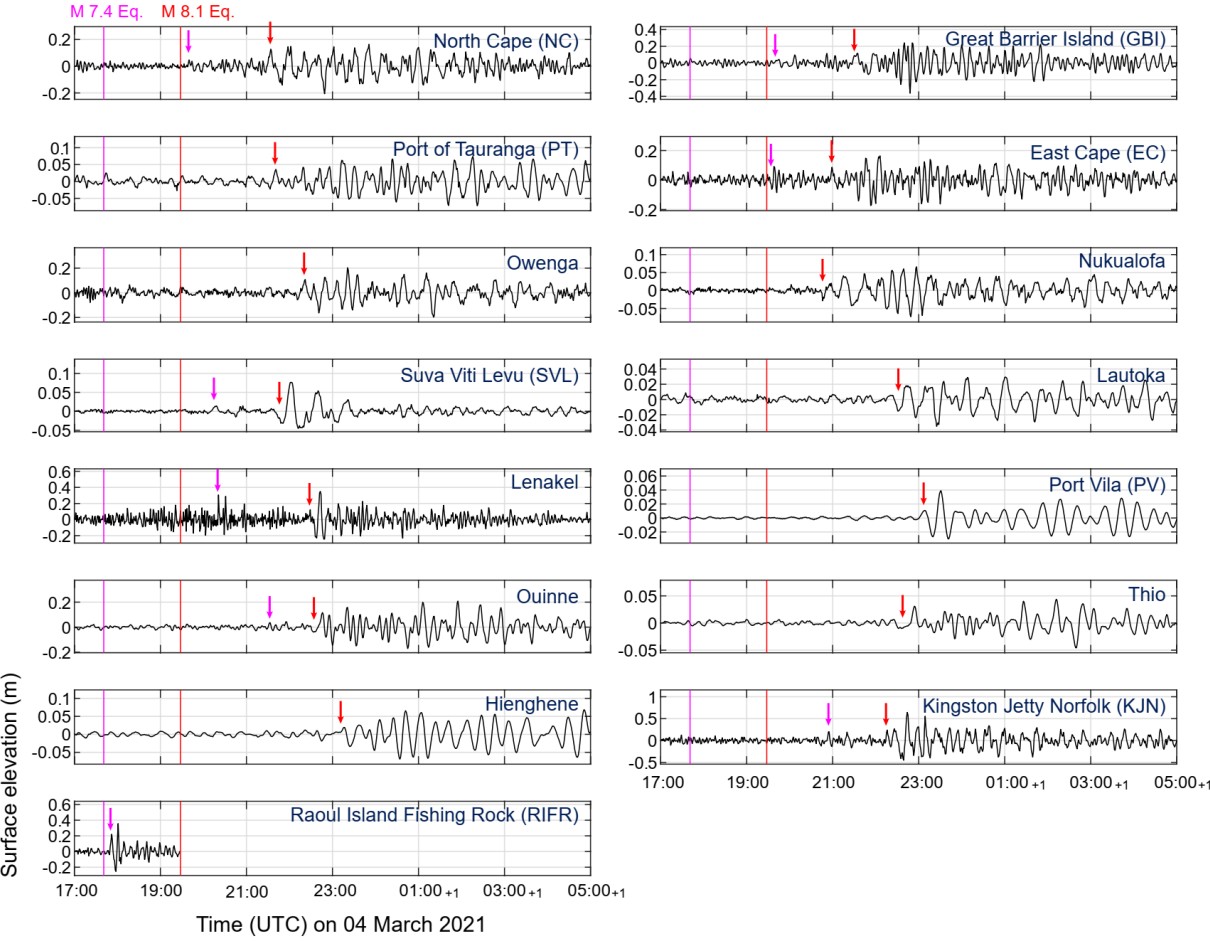

**Figure 2: Sea level change recorded by tide gauges showing two tsunamis. The pink and red vertical lines show the origin times of the M_w 7.4 and M_w 8.1 earthquakes, respectively. The arrival times of the first and second tsunamis are marked by pink and red arrows, respectively.**

### 2.4 Calculating Tsunami Source Period

In this study, we calculated tsunami source period from finite fault models to compare with the results of spectral analyses. Theoretically, the tsunami source period is related to earthquake rupture length and water depth (Rabinovich, 1997; 2010; Heidarzadeh and Satake, 2013; Wang et al., 2021). It can be estimated as:

$$T_n = \frac{2L}{n\sqrt{gh}} \quad n = 1,2,3,\dots \; , \tag{1}$$

where $L$ is the typical size of tsunami source area (length or width), $g$ is the gravity acceleration, and $h$ is the average water depth in source area.

## 3 Tsunami Waveform Analysis

Two tsunamis arrive successively at 15 tide gauges (Figure 2). The first tsunami arrived at the RIFR station 11 min after the foreshock. The first peak and the maximum amplitudes are 0.21 m and 0.34 m, respectively. It reached EC at 117 min after the earthquake with a maximum amplitude of 0.09 m. It is noteworthy that the Lenakel tide gauge, located in the direction parallel to the short axis of the earthquake source fault, recorded a maximum amplitude of 0.31 m. It is the largest value among all stations except for the RIFR. The first tsunami was also recorded by KJN with a maximum amplitude of 0.21 m.

The arrival times of the second tsunami are generally consistent with the results of the TTT calculations (Figure 1; dashed contours) with clear large amplitude signals. It arrived at EC at 20:59 (UTC); 91 min after the mainshock with an amplitude of 0.09 m. Yet the maximum amplitude at this station (0.17 m) appeared in a later time (approximately one hour after the first arrival) (Figure 2). The largest waves were also observed in later time in other stations such as NC, GBI, Nukualofa, Owenga, Ouinne, and Hienghene (Figure 2). Such late arrival of largest tsunami wave was reported for other tsunamis in the past such as for the 2004 Indian Ocean tsunami (Rabinovich and Thomson, 2007) and the 2020 Aegean Sea tsunami (Heidarzadeh et al., 2021). The largest tsunami amplitude (0.64 m) was recorded by KJN, which is also located in the direction parallel to the short axis of the fault. Besides, the second tsunami has a longer wavelength than the first tsunami, leading to longer tsunami periods. At stations without clear signal of the first tsunami (e.g., Lautoka), the tsunami waveforms are dominant by the long-period components which are generated by the second tsunami. However, the stations that recorded both events show the superposition of two successive tsunamis. For example, the waveform of the second tsunami at EC is mixed up with the short-period components in the few hours after its arrival (21:00–24:00), likely caused by the oscillation of the first tsunami (Figure 2). After 24:00, the tsunami waveforms mainly present long-period components, possibly due to the dissipation of the short-period waves of the first tsunami. Similar patterns were also observed at Ouinne: Short-period components existed in the few hours after the second tsunami's arrival, but the waveforms after 01:00 (+1) were dominated by long-period components (Figure 2).

## 4 Spectral Analyses

Figures 3 presents the results of Fourier analyses for two tsunamis and background signals. The gap between the tsunami and background spectra is attributed to the tsunami energy. The periods of the tsunami spectral peaks generally contain the effects of tsunami source, propagation path, and local topography. According to Figure 3, the second tsunami has larger energy (spectral power) than the first tsunami due to the larger magnitude of the second earthquake. The background spectra are smoother than tsunami spectra at most stations. The peak periods of the first tsunami are mostly distributed in the range of 5–17 min, whereas the dominant periods range for the second tsunami is approximately 8–32 min. In other words, the second tsunami has generated longer-period waves, which is natural as the source dimension of the second earthquake

(240 km × 190 km) is much larger than that of the first earthquake (80 km × 80 km). In Figure S1 of Supporting Information, we plotted the Fourier spectra of DART stations NZG and NZI. The spectral peaks of these stations were generally consistent with those of tide gauges. Regarding tsunami dominant period (or peak periods), here we chose the period range

that more than half of the stations present as the dominant range. In Table 1, we listed the peak periods at each tide gauge for the two tsunamis. Tsunami spectra can help identify the source size, potential asperities and other information about earthquake source processes. The dominant wave period ranges for tsunami events are related to the size of the source, which we explain in Section 5. Assuming the same water depths, tsunamis generated by earthquakes with larger source sizes normally have longer dominant wave periods. For example, the tsunami generated by the Mw 9.1 2004 Sumatra earthquake

in the near-field region indicated dominance of long waves with periods of 30–60 min (Rabinovich et al., 2006).

**Table 1: Peak periods at each tide gauge for two tsunami events. The values were calculated by Fourier analyses. Station name abbreviations are: North Cape (NC), Great Barrier Island (GBI), East Cape (EC), Suva Viti Levu (SVL), Kingston Jetty Norfolk (KJN), Port Vila (PV), and Raoul Island Fishing Rock (RIFR).**

| Tide gauge | Peak period(s) for the first tsunami (min) | Peak period(s) for the second tsunami (min) |
|---|---|---|
| NC | 9.1 | 9.8; 21.3 |
| GBI | 6.5; 10.7 | 6.4; 10.7; 32.0 |
| PT | N/A | 9.8 |
| EC | 6.1; 9.8; 16.0 | 8.5; 18.3 |
| Owenga | N/A | 14.2 |
| Nukualofa | N/A | 7.1; 21.3 |
| SVL | 8.0; 18.3 | 32.0 |
| Lautoka | N/A | 9.8; 25.6 |
| Lenakel | 6.1; 8.5 | 5.6; 12.8 |
| PV | N/A | 25.6 |
| Ouinne | 8.5; 25.6 | 9.1; 32.0 |
| Thio | N/A | 8.5; 14.2 |
| Hienghene | N/A | 7.5; 18.3 |
| KJN | 8.5; 14.2 | 14.2 |
| RIFR | 4.6; 9.1 | N/A |

Wavelet analyses reveal the variations of dominant tsunami frequency over time (Figure 4). At EC, Lenakel, and KJN, the arrivals of two successive tsunamis can be clearly identified on the wavelet plots. The second tsunami has larger energy levels and longer-period waves than the first tsunami (Figure 4), which is consistent with the results of Fourier analyses. At other stations such as GBI, the arrival time of the first tsunami is not clear. At most tide gauges, there are two oscillation

patterns visible on the wavelet plots. One oscillation pattern has dominant periods in the range of 5–17 min. The other one
has dominant periods over 15 min and up to approximately 30 min, which occurs after the first pattern. These two period
bands on the wavelet plots reflect the first and the second tsunamis. For example, the wavelet plot of EC reveals a persisting
wave in the period range of 5–11 min. The oscillation begins at approximately 2 h after the $M_w$ 7.4 earthquake, which
corresponds to the arrival time of the first tsunami. Another oscillation with longer dominant periods begins at approximately
2 h after the $M_w$ 8.1 earthquake, which is consistent with the arrival time of the second tsunami. The oscillation lasts for
more than 5 h. It is noteworthy that after the arrival of the second tsunami, the oscillation pattern with short dominant
periods still exists and becomes even stronger sometimes, indicating the contribution of both tsunamis at EC. Two oscillation
patterns simultaneously exist for several hours and almost simultaneously diminish. The wavelet plot of KJN shows similar
patterns to that of EC. Although there are also two oscillation patterns on the wavelet plot of Lenakel, the long-period
oscillation only lasts for less than 5 h. To the contrary, at GBI, the oscillation pattern with long dominant periods lasts for
approximately 3 h, whereas the oscillation pattern with short dominant periods lasts for more than 6 h. At Ouinne, a
persisting wave in the period band of 20–30 min is visible with high energy. Another oscillation pattern (6–12 min)
diminishes earlier, which explains the reason why the tsunami waveforms at Ouinne has less short-period components in the
later phases (Figure 2). Nevertheless, different from other stations, we cannot find a long-lasting wave at SVL. There is only
one oscillation period band (20–40 min) and it diminishes rapidly. We note that the dispersive effects of tsunamis from the
second event are evident on the wavelet plots as tsunami dominant period for the few initial waves is around ~20 min,
whereas it linearly shifts towards ~10 min for the later waves, giving us the opportunity to plot the inverse dispersion lines
(black dashed lines in Figure 4). We plotted the dispersion curve on these diagrams. We also observe short-period waves
with period of 5–8 min at some sea-level stations (Table 1; Figures 3–4), which we attribute to various local bathymetric
effects.  In addition, we also note that wavelets and Fourier analyses give spectral results with varying degrees of accuracies,
because wavelet analysis also incorporates the time evolution and thus its spectra are not usually as detailed as those
obtained by Fourier analyses.

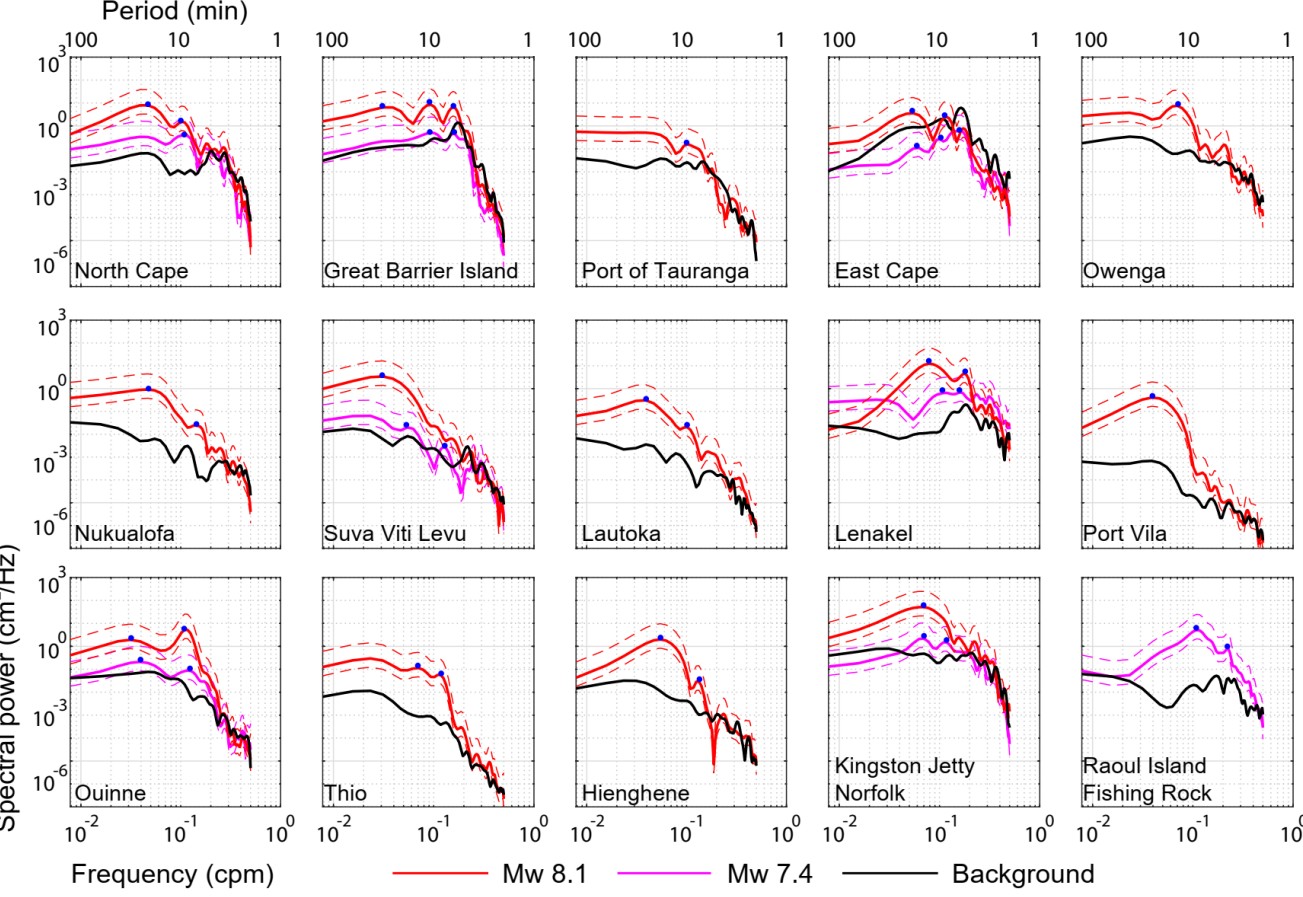

**Figure 3: Fourier analyses for tsunamis generated by two successive earthquakes ($M_w$ 7.4 and $M_w$ 8.1) in Kermadec Islands. Pink and red curves represent the spectra of the first tsunami and the second tsunami, respectively. Blue dots show the spectral peaks listed in Table 1. The 95% confidence bounds of two tsunami spectra are indicated by dashed curves. The background spectra (black curves) are also plotted for comparison.**

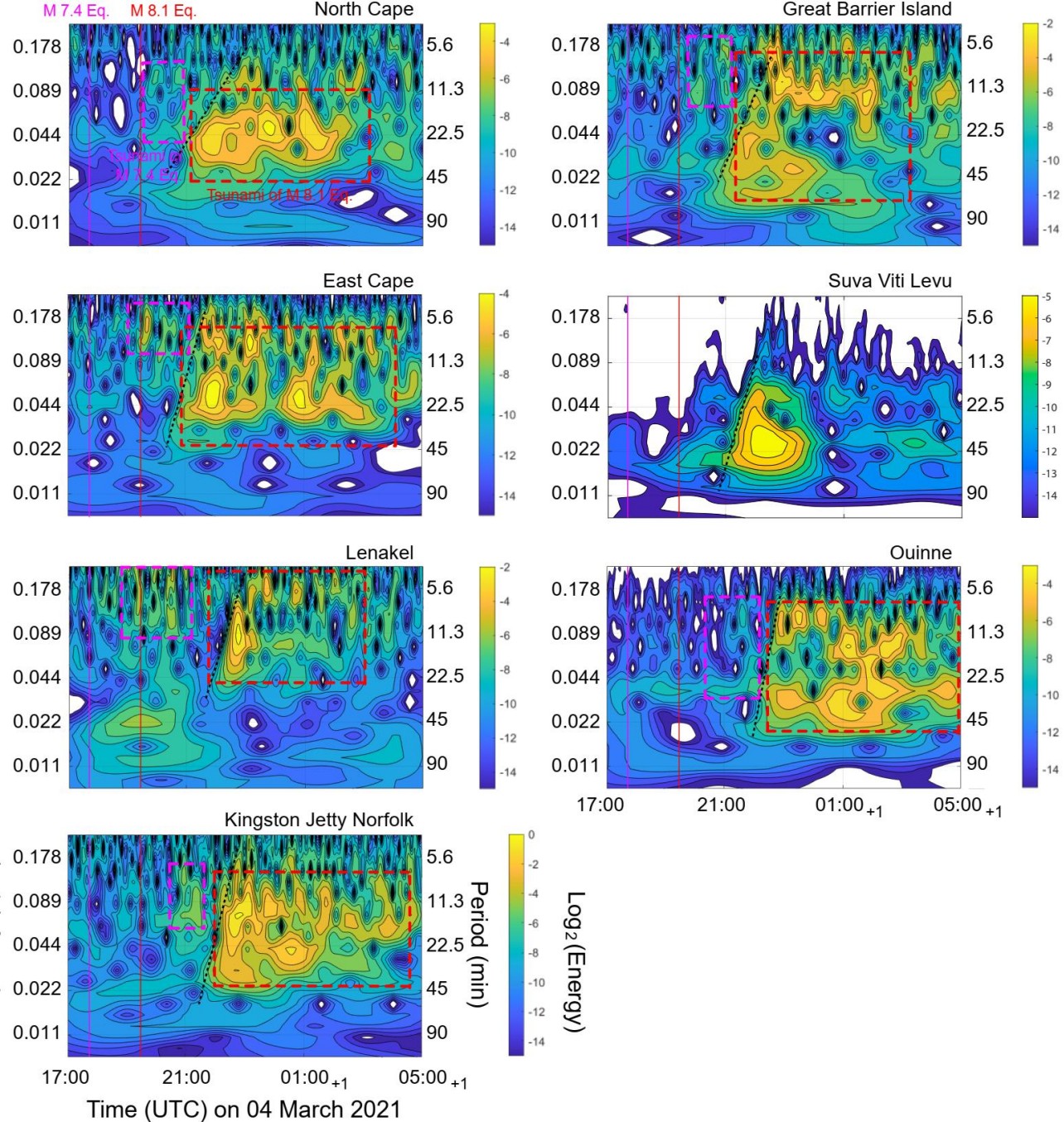

**Figure 4: Wavelet (frequency-time) analyses for tsunamis generated by two successive earthquakes ($M_w$ 7.4 and $M_w$ 8.1) in Kermadec Islands. The colormap shows levels of spectral energy at different times and periods. For guidance, we marked the dominant periods of two tsunamis by pink ($M_w$ 7.4) and red ($M_w$ 8.1) rectangles. The pink and red vertical lines show the origin times of the $M_w$ 7.4 and $M_w$ 8.1 earthquakes, respectively. The dispersion curves are plotted by black dashed lines. On the horizontal axis, plus one (+1) indicates one day passed.**

## 5 Reconstructing the Tsunami Source Spectrum

A tsunami source spectrum reveals the earthquake source characteristics without the effects of tsunami propagation path or local topography. In our study, the epicenters of two earthquakes are close to each other (~55 km; Figure 1); the earthquakes are of similar mechanism (both thrust events). We simulated the propagation of two tsunamis using *JAGURS* code and plotted their maximum amplitude in our region of interest to investigate their propagation path (Figures 5a and 5b). The tsunami amplitude in the NW-SE direction is larger than that in the NE-SW direction because it is parallel to the short axis of the fault. The propagation paths of two successive tsunamis are similar. Hence, it enables us to reconstruct tsunami source spectrum using EGF method which assumes that the smaller event acts as the EGF for the larger event (Miller, 1972; Heidarzadeh et al., 2016). We computed the spectral ratio of the second tsunami to the first tsunami at seven tide gauges (Figure 5c; gray curves), and then calculated their normalized average values (Figure 5c; blue curve) through adjusting the peak energy of all spectra to that of the largest one. The source spectrum shows that the energy of the second tsunami is mainly distributed in the period range of 8–30 min, with spectral peaks at 25.6 min, 16.0 min, and 9.8 min. The period range is generally consistent with the results of Fourier analyses of each station for the second event (Figure 3). Figure 5d shows the results of the method proposed by Rabinovich (1997) which is based on dividing the tsunami spectra to that of the background to construct tsunami source spectrum. We computed the spectral ratio of the second tsunami to the background signals at all tide gauges except RIFR and calculated their normalized average. The period range of main energy (7–28 min) contains spectral peaks at 25.6 min, 16.0 min and 8.5 min, which are close to the spectral peaks calculated by EGF method. In addition, we also computed the spectral ratio of the first tsunami to the background signals at those tide gauges with evident records and calculated their normalized average (Figure 5e). This plot yields only the dominant periods of the first tsunami (generated by Mw 7.4 earthquake) showing that the energy is mainly distributed in the period range of 5–17 min, indicating that the size of the tsunami source of the first event is smaller than that of the second event.

According to the USGS model, the size of the $M_w$ 8.1 earthquake is 240 km (length of the fault) × 190 km (width of the fault). However, the non-zero displacement region is approximately 210 km × 170 km (https://earthquake.usgs.gov/earthquakes/eventpage/us7000dflf/finite-fault). The average water depth in the source area is ~ 5,000 m. Hence, the first three source periods of the short axis of the source (width) using the analytical equation (Eq. 1) are calculated as 25.6 min, 12.8 min, and 8.5 min. These values are consistent with the results of spectral analyses of the observed waves based on the EGF and tsunami-to-background spectral ratio methods (Figures 5c; 5d) showing peak periods at 25.6 min, 16.0 min and 9.8 min (8.5 min). We acknowledge that Equation (1) is a rough approximation of dominant tsunami source periods, and therefore we allowed a discrepancy of up to 20% while making the comparison. We note that the periods of tsunami waves are mainly influenced by the short axis (width) (Heidarzadeh and Satake, 2013).

In addition, the results of two methods (i.e., EGF and tsunami-to-background spectral ratio) show similarities in shapes and peak periods of tsunami source spectrum (Figures 5c and 5d). It is noted that the EGF method has the capability to

remove both propagation-path effects and local bathymetric effects from the tide gauge records whereas the tsunami-to-background spectral ratio method would remove mainly local bathymetric effects. Hence, our results may imply that the effects of propagation path are negligibly small for this case, where the tide gauges are located at distances between ~100 km and ~ 2,000 km from the source. Both of the methods are able to reveal the source characteristics merely based on tsunami observations rather than seismological data.

As limitations of this study, we could mention a few items: We are not using DART data (Figure S1) to compute tsunami source spectrum due to the short duration of high-sampling records. In general, DART records are valuable type of sea level data in terms of tsunami source studies because they are less affected by local and regional bathymetry. In addition, the number of sea level records that we used for analyses of these tsunamis is not very large due to the limited number of available stations.

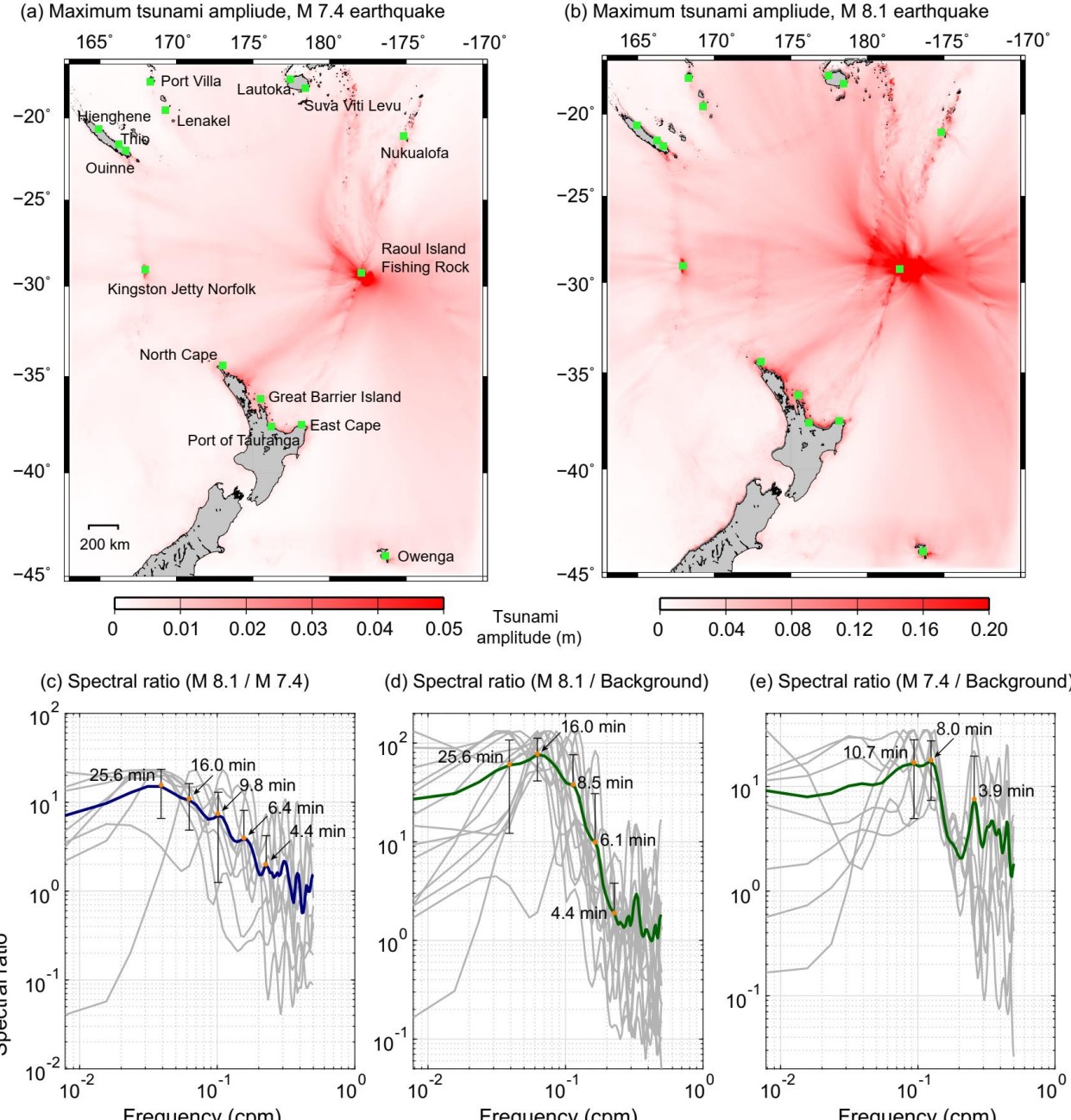

**Figure 5:** **(a, b) Maximum simulated amplitudes for two tsunamis during the entire simulation time. The source models used for numerical simulation are from the USGS. (c) Spectral ratio of two tsunamis by dividing the spectral energy of the second tsunami to that of the first tsunami (EGF method). Blue curve is the normalized average of tsunami spectral energy at different tide gauges. (d) Spectral ratio of the second tsunami spectrum to the background signal spectrum. Green curve is the normalized average of**

different tide gauges. (e) Spectral ratio of the first tsunami spectrum to the background signal spectrum. Green curve is the normalized average of different tide gauges.

## 6 Conclusion

We studied the characteristics of the tsunamis generated by two earthquakes ($M_w$ 7.4 and $M_w$ 8.1) that occurred in the Kermadec Islands successively on March 4, 2021, within ~55 km from each other and within an approximately 2 h interval. We used the sea level records of 15 tide gauges. The spectra of Fourier analyses show that the dominant period bands of the first tsunami and the second tsunami are 5–17 min and 8–32 min, respectively. Two oscillation patterns with different period ranges are visible on the wavelet plots at most stations which belong to the first and the second tsunamis. We observed that after the arrival of the second tsunami, the oscillation in the period range of the first tsunami still exists. We calculated the

tsunami source spectrum of the larger event (i.e., the second tsunami) by two approaches: empirical Green's function (EGF) method and tsunami-to-background ratio method. Using the first tsunami as the EGF, spectral deconvolution indicated that energy of the second tsunami is mainly distributed in the period range of 8–30 min, with spectral peaks at 25.6 min, 16.0 min, and 9.8 min. The method of tsunami-to-background ratio yielded similar results to the EGF method. The source characteristics were obtained merely based on tsunami data and thus these two methods could be complementary to

seismological approaches in source analysis.

## Data availability

The datasets were derived from sources in the public domain. The tide gauge data used in this research were provided by the Sea Level Station Monitoring Facility of the Intergovernmental Oceanographic Commission (http://www.ioc-sealevelmonitoring.org/list.php). We used bathymetric and topographic data of the General Bathymetric Chart of the Ocean

(https://www.gebco.net/data_and_products/gridded_bathymetry_data/). We used earthquake focal mechanism catalog of the Global Centroid Moment Tensor Project (https://www.globalcmt.org/CMTsearch.html) and earthquake source model of the United States Geological Survey (https://earthquake.usgs.gov/earthquakes/eventpage/us7000dflf/executive). We used the GMT software for drafting some of the figures (Wessel and Smith, 1998).

## Author contribution

YW is responsible for methodology, data curation, software, and writing - original draft preparation. MH is responsible for methodology, software, and writing- reviewing and editing. KS is responsible for methodology, supervision, and writing-reviewing and editing. GH is responsible for data curation and software.

**Competing Interests**

The authors declare that they have no conflict of interest.

**Acknowledgements**

The authors declare that they have no conflict of interest. We thank Dr. Takane Hori and Dr. Kentaro Imai from Japan Agency for Marine-Earth Science Technology for their valuable suggestions. MH is funded by the Royal Society (the United Kingdom), grant number CHL\R1\180173. YW is funded by Japan Society for the Promotion of Science, grant number 19J20293.

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
