# Peer review of "Characteristics of two tsunamis generated by successive $M_w$ 7.4 and $M_w$ 8.1 earthquakes in Kermadec Islands on March 4, 2021"

_Natural Hazards and Earth System Sciences, 2021_

## Referee Comment (RC1)

Review

of NHESS-2021-369 "*Characteristics of two tsunamis generated by successive Mw 7.4 and Mw 8.1 earthquakes in Kermadec Islands on March 4, 2021*" by Yuchen Wang, Mohammad Heidarzadeh, Kenji Satake, and Gui Hu

This is an interesting and, in general, is well written paper. However, I believe that some of the results can be and should be presented in a more informative and spectacular way and the entire text and figures should be polished! Therefore, my recommendation "Acceptable after revision".

(1) My main suggestion is related to Figure 4 and the respective text. The most noteworthy point of this event and the corresponding study is that there were TWO earthquakes with an interval less than 2 hours between them. The authors declare that the tsunami waves associated with both events are clearly seen in Figure 4. But for readers (in contrast to the authors) this is not obvious! What the authors should do is to construct the theoretical *dispersion curves* for both events (following papers of *González and Kulikov*, 1993 and *Kulikov,* 2006) and demonstrate that the observed *f-t* extrema are in good agreement with the theory. BTW, the source regions of these two events are relatively small; therefore, the dispersive effects for propagating waves should be clearly seen in the wavelet plots. All details and necessary equations can be found in *Fine et al.* (2019).

The dispersion curves and corresponding *f-t* diagrams are constructed as functions of *frequency* (not of period!) (e.g. *Thomson and Emery,* 2014); now all plots in Figure 4 are upside down and wave dispersion is difficult to identify. (Actually, the authors themselves call these plots "frequency-time analyses", not "period-time"). So the figures should be presented in the standard way.

I believe that if the authors prepare everything in the best way, the corresponding figure will become striking and highly quoted! The authors of the listed papers used the *f-t* diagrams to identify a specific event and to examine the dispersive properties of tsunami waves. However, I do not know any study where this approach has been used to identify and separate two events following each other!

(2) Spectral analysis, Figure 3. This figure is prepared in a very "unfriendly manner"! I know that the second author loves to combine a numerous number of plots in one figure. However, in that case the value of each plot tends to zero. What are the specific spectral features that the authors would like to demonstrate? Spectral peaks? The differences of the tsunami spectra from the background spectra? The differences between two tsunamis? Any of these features are unclear in this figure.

The spectra are strongly vertically compressed; the scale of the Y-axis looks strange: $10^{-5}$, then $10^0$ and nothing between! As a result, all spectra look flat. BTW, I believe that the dimension of the Y-axis is $cm^2/Hz$, not $cm/Hz$.

An additional question is units. Typical periods of seismic waves are second; therefore, it is natural use Hz for their frequencies. However, typical periods of tsunami waves are minutes and fractures of an hour. Besides, the sampling interval of the data is 1 min. Thus, it would be natural for tsunami spectra to use cycles per minute (cpm) or cycles per hour (cph). This will be helpful for readers, and they will not need to use a calculator to estimate the period of a specific spectral peak. (The same comment is just for Figures 5c and 5d).

Also, what is the meaning of the magic numbers for periods: 1,7, 16.7 and 166.7 min in the scale of periods?! The Nyquist period for these spectra is 2 min, the fundamental period is 120 min (2 hours). The meaning of the shown magic periods remains totally unclear and does not help a reader to detect the periods of spectral maxima.

The last but not least comment to these spectra: the confidence levels are not shown and without confidence levels all results of spectral analysis are senseless (e.g. *Thomson and Emery*, 2014).

(3) Figure 5c. The idea to estimate "relative spectra of tsunamis" as the ratio of various tsunami events was first proposed by *Miller* (1972). The authors mention this study (Line 44), but it is absent in the List of references. Also, they do not mention this paper when they discuss their Figure 5c.

It should be emphasized that this approach (the authors call it "empirical Green's function", EGF) does not allow to reconstruct the tsunami source spectrum because it shows not the properties of the second source themselves, but the *differences* of the second source from the first one. As was indicated by the authors (e.g. Lines 168-169), the seismic mechanisms of the two events are very similar. Therefore, the mutual part of the two sources (in particular, mutual spectral peaks) is not seen in Figure 5c. From this point of view, it would be interesting to see the tsunami/background ratio for the first event (i.e. a figure similar to Figure 5d, but for the first tsunami).

(4) Equation (1) (Line 183) is the exact solution for periods of standing (eigen) modes in a closed rectangular basin of uniform depth $h = const$. This equation for $n = 1$ allows to estimate very roughly the *order* of periods of generated tsunami waves. However, a real tsunami source is far away from being uniform and rectangular; thus, even for the first mode, this estimate is a very approximate. So, this estimate is rather qualitative than quantitative. From this point of view, it is strange to see that the authors use this equation for PRECISE estimation (with fractures of minutes!) the "source periods", and even not only for the first but also for the secondary modes.

(5) The authors use a high-pass filter to remove tides from the original records; this is definitely not the best way to suppress tides! Any unnoticed spike, shift or gap (small in comparison with tides) will strongly distort the tsunami signal (and even create some "artificial tsunamis"; the corresponding examples are well known!). It is much better to subtract predicted or calculated tides (as was done by the authors in some other their papers. BTW, the authors write: "*…we applied a second-order high-pass filter with the corner frequency of 0.00014 Hz (7,200 s) to remove the tidal components*". Why "Hz", why "seconds"?! The sampling interval of the data is not seconds, but 1 min; it would be much easier for readers if the authors write: "We applied a 2-hour (or 120-min) high-pass window"!

Some polishing of the paper language will be useful; there are some problems with articles, etc.

Alexander Rabinovich

References

Fine, I. V., Thomson, R. E., Chadwick, W. W., Jr, and Fox, C. G. (2019), Toward a universal frequency of occurrence distribution for tsunamis: statistical analysis of a 32-year bottom pressure record at Axial Seamount, *Geophysical Research Letters,* 47, e2020GL087372. https://doi.org/10.1029/2020GL087372

González, F. I., & Kulikov, E. A. (1993). Tsunami dispersion observed in the deep ocean. In S. Tinti (Ed.), *Tsunamis in the World* (pp. 7–16). Dordrecht: Kluwer Acad. Publ.

Kulikov, E. (2006), Dispersion of the Sumatra Tsunami waves in the Indian Ocean detected by satellite altimetry, *Russian Journal of Earth Sciences*, 8, ES4004, doi:10.2205/2006ES000214.

Miller, G.R. (1972), Relative spectra of tsunamis. *Publ. HIG-72-8*, Hawaii Institute of Geophysics, University of Hawaii, Honolulu, 7 pp.

Thomson, R. E., & Emery, W. J. (2014). *Data Analysis Methods in Physical Oceanography* (3rd ed., p. 716). New York: Elsevier Science.

---

## Author Comment (AC1)

**Response to RC1**

7th January 2022

Dear Dr. Rabinovich,

We sincerely thank you for the constructive comments that greatly helped us to improve the manuscript. Here we present our point-by-point responses and revision to the comments.

Sincerely,

Yuchen Wang
Postdoctoral Researcher,
Japan Agency for Marine-Earth Science and Technology,
Japan
Email: ywang@jamstec.go.jp

**General Review Comments**

*This is an interesting and, in general, is well written paper. However, I believe that some of the results can be and should be presented in a more informative and spectacular way and the entire text and figures should be polished! Therefore, my recommendation "Acceptable after revision".*

**Comment 1**

*My main suggestion is related to Figure 4 and the respective text. The most noteworthy point of this event and the corresponding study is that there were TWO earthquakes with an interval less than 2 hours between them. The authors declare that the tsunami waves associated with both events are clearly seen in Figure 4. But for readers (in contrast to the authors) this is not obvious! What the authors should do is to construct the theoretical dispersion curves for both events (following papers of González and Kulikov, 1993 and Kulikov, 2006) and demonstrate that the observed f-t extrema are in good agreement with the theory. BTW, the source regions of these two events are relatively small; therefore, the dispersive effects for propagating waves should be clearly seen in the wavelet plots. All details and necessary equations can be found in Fine et al. (2019).*

*The dispersion curves and corresponding f-t diagrams are constructed as functions of frequency (not of period!) (e.g. Thomson and Emery, 2014); now all plots in Figure 4 are upside down and wave dispersion is difficult to identify. (Actually, the authors themselves call these plots "frequency-time analyses", not "period-time"). So the figures should be presented in the standard way.*

*I believe that if the authors prepare everything in the best way, the corresponding figure will become striking and highly quoted! The authors of the listed papers used the f-t diagrams to identify a specific event and to examine the dispersive properties of tsunami waves. However, I do not know any study where this approach has been used to identify and separate two events following each other!*

**Response and Revision**

Thank you. This is very helpful. In response to this comment, we revised Figure 4.

(1) We plotted the dispersion curves at those tide gauges whose dispersive effects are clear. In Line 172 (Please note that the number of line refers to the revised file with track changes), we added:

'We note that the dispersive effects of tsunamis from the second event are evident on the wavelet plots as tsunami dominant period for the few initial waves is around ~20 min, whereas it linearly shifts towards ~10 min for the later waves, giving us the opportunity to plot the inverse dispersion lines (black dashed lines in Figure 4).'

In the caption of Figure 4, we added:

'The dispersion curves are plotted by black dashed lines.'

(2) We used f-t diagrams for Figure 4 rather than period-time diagrams. In Line 153, we changed:

'Wavelet analyses reveal the variations of dominant tsunami  frequency over time (Figure 4).'

**Comment 2**

*Spectral analysis, Figure 3. This figure is prepared in a very "unfriendly manner"! I know that the second author loves to combine a numerous number of plots in one figure. However, in that case the value of each plot tends to zero. What are the specific spectral features that the authors would like to demonstrate? Spectral peaks? The differences of the tsunami spectra from the background spectra? The differences between two tsunamis? Any of these features are unclear in this figure.*

*The spectra are strongly vertically compressed; the scale of the Y-axis looks strange: 10-5 , then 100 and nothing between! As a result, all spectra look flat. BTW, I believe that the dimension of the Y-axis is cm2 /Hz, not cm/Hz.*

*An additional question is units. Typical periods of seismic waves are second; therefore, it is natural use Hz for their frequencies. However, typical periods of tsunami waves are minutes and fractures of an hour. Besides, the sampling interval of the data is 1 min. Thus, it would be natural for tsunami spectra to use cycles per minute (cpm) or cycles per hour (cph). This will be helpful for readers, and they will not need to use a calculator to estimate the period of a specific spectral peak. (The same comment is just for Figures 5c and 5d).*

*Also, what is the meaning of the magic numbers for periods: 1,7, 16.7 and 166.7 min in the scale of periods?! The Nyquist period for these spectra is 2 min, the fundamental period is 120 min (2 hours). The meaning of the shown magic periods remains totally unclear and does not help a reader to detect the periods of spectral maxima.*

*The last but not least comment to these spectra: the confidence levels are not shown and without confidence levels all results of spectral analysis are senseless (e.g. Thomson and Emery, 2014).*

**Response and Revision**

Acknowledged. We made several changes to address this comment.
(1) We plotted these spectra to show the main energy distribution of two tsunamis at each tide gauge. The gap between the tsunami and background spectra is attributed to the tsunami energy. In response to this comment, we added notations of main spectral peaks of two tsunamis to Figure 3 and listed these values in Table 1.
(2) We changed the unit of y-axis from to cm/Hz to $cm^2$/Hz. In addition, we also made the y-axis less compressed to show the vertical contents clearly.
(3) We changed the unit of x-axis from Hz to cycles per minute (cpm). We also made such changes to Figures 5c and 5d.
(4) We changed the annotation of periods to 100, 10, and 1 min.
(5) We added the 95% confidence bounds to these spectra. In the caption of Figure 3, we added:

'The 95% confidence bounds of two tsunami spectra are indicated by dashed curves.'

**Comment 3**

*Figure 5c. The idea to estimate "relative spectra of tsunamis" as the ratio of various tsunami events was first proposed by Miller (1972). The authors mention this study (Line 44), but it is absent in the List of references. Also, they do not mention this paper when they discuss their Figure 5c.*

*It should be emphasized that this approach (the authors call it "empirical Green's function", EGF) does not allow to reconstruct the tsunami source spectrum because it shows not the properties of the second source themselves, but the differences of the second source from the first one. As was indicated by the authors (e.g. Lines 168-169), the seismic mechanisms of the two events are very similar. Therefore, the mutual part of the two sources (in particular, mutual spectral peaks) is not seen in Figure 5c. From this point of view, it would be interesting to see the tsunami/background ratio for the first event (i.e. a figure similar to Figure 5d, but for the first tsunami).*

**Response and Revision**

In response to this comment, we added Miller (1972) to the Reference. We also plotted the tsunami/background ratio for the first event in Figure 5e.

(1) In Line 194, we added:

'…assumes that the smaller event acts as the EGF for the larger event (Miller, 1972; Heidarzadeh et al., 2016)'.

(2) In Line 203, we added:

'In addition, we also computed the spectral ratio of the first tsunami to the background signals at those tide gauges with evident records and calculated their normalized average (Figure 5e). This plot yields only the dominant periods of the first tsunami (generated by Mw 7.4 earthquake) showing that the energy is mainly distributed in the period range of 5–17 min, indicating that the size of the tsunami source of the first event is smaller than that of the second event.'

(3) In the caption of Figure 5, we added:

'(e) Spectral ratio of the first tsunami spectrum to the background signal spectrum. Green curve is the normalized average of different tide gauges.'

**Comment 4**

*Equation (1) (Line 183) is the exact solution for periods of standing (eigen) modes in a closed rectangular basin of uniform depth h = const. This equation for n = 1 allows to estimate very roughly the order of periods of generated tsunami waves. However, a real tsunami source is far away from being uniform and rectangular; thus, even for the first mode, this estimate is a very approximate. So, this estimate is rather qualitative than quantitative. From this point of view, it is strange to see that the authors use this equation for PRECISE estimation (with fractures of minutes!) the "source periods", and even not only for the first but also for the secondary modes.*

**Response and Revision**

We agree. We referred to Rabinovich (2010) for Equation (1). We acknowledge that this equation is a rough estimation of the period. In this paper, though we calculated the source periods with a fraction of minutes, we still regarded it as an approximation. A discrepancy of up to 20% was allowed when we make the comparison. In response to this comment, we added more explanations.

(1) In Line 214, we added:

"We acknowledge that Equation (1) is a rough approximation of dominant tsunami source periods, and therefore we allowed a discrepancy of up to 20% while making the comparison."

**Comment 5**

*The authors use a high-pass filter to remove tides from the original records; this is definitely not the best way to suppress tides! Any unnoticed spike, shift or gap (small*

*in comparison with tides) will strongly distort the tsunami signal (and even create some "artificial tsunamis"; the corresponding examples are well known!). It is much better to subtract predicted or calculated tides (as was done by the authors in some other their papers. BTW, the authors write: "...we applied a second-order high-pass filter with the corner frequency of 0.00014 Hz (7,200 s) to remove the tidal components". Why "Hz", why "seconds"?! The sampling interval of the data is not seconds, but 1 min; it would be much easier for readers if the authors write: "We applied a 2-hour (or 120-min) high-pass window"!*

**Response and Revision**

We agree. We acknowledge that subtracting tides from the original records is a good way to suppress tides. Here we refer to Heidarzadeh et al. (2015 *PAAG*), which showed that the high-pass filtering yields similar results as subtracting tides, because tsunami is a very long wave. In our study, we conducted quality control to ensure that there were no spike, shift, or gap in the time series that we analyzed. In response to this comment, we explained our method and changed the expression.

(1) In Line 68, we changed:

"Then, we applied a  2-hour (120-min) high-pass filter to remove the tidal components (Figure 2) (Heidarzadeh and Satake, 2013)."

(2) In Line 70, we added:

"Heidarzadeh et al. (2015) showed that the high-pass filtering yields similar results as subtracting calculated tides from the original records."

---

## Author Comment (AC2)

[Figure]

**Figure 3:** **Fourier analyses for tsunamis generated by two successive earthquakes ($M_w$ 7.4 and $M_w$ 8.1) in Kermadec Islands. Pink and red curves represent the spectra of the first tsunami and the second tsunami, respectively. Green dots show the spectral peaks listed in Table 1. The 95% confidence bounds of two tsunami spectra are indicated by dashed curves. The background spectra (black curves) are also plotted for comparison.**

[Figure]

**Figure 4: Wavelet (frequency-time) analyses for tsunamis generated by two successive earthquakes (M_w 7.4 and M_w 8.1) in Kermadec Islands. The colormap shows levels of spectral energy at different times and periods. For guidance, we marked the dominant periods of two tsunamis by pink (M_w 7.4) and red (M_w 8.1) rectangles . The pink and red vertical lines show the origin times of the M_w 7.4 and M_w 8.1 earthquakes, respectively. The dispersion curves are plotted by black dashed lines. On the horizontal axis, plus one (+1) indicates one day passed.**

[Figure]

Figure 5: (a, b) Maximum simulated amplitudes for two tsunamis during the entire simulation time. The source models used for numerical simulation are from the USGS. (c) Spectral ratio of two tsunamis by dividing the spectral energy of the second tsunami

to that of the first tsunami (EGF method). Blue curve is the normalized average of tsunami spectral energy at different tide gauges. (d) Spectral ratio of the second tsunami spectrum to the background signal spectrum. Green curve is the normalized average of different tide gauges. (e) Spectral ratio of the first tsunami spectrum to the background signal spectrum. Green curve is the normalized average of different tide gauges.

20

**Table 1: Peak periods at each tide gauge for two tsunami events. The values were calculated by Fourier analyses.**

**Station name abbreviations are: North Cape (NC), Great Barrier Island (GBI), East Cape (EC), Suva Viti Levu (SVL), Kingston Jetty Norfolk (KJN), Port Vila (PV), and Raoul Island Fishing Rock (RIFR).**

| Tide gauge | Peak period(s) for the first tsunami (min) | Peak period(s) for the second tsunami (min) |
|---|---|---|
| NC | 9.1 | 9.8; 21.3 |
| GBI | 6.5; 10.7 | 6.4; 10.7; 32.0 |
| PT | N/A | 9.8 |
| EC | 6.1; 9.8; 16.0 | 8.5; 18.3 |
| Owenga | N/A | 14.2 |
| Nukualofa | N/A | 7.1; 21.3 |
| SVL | 8.0; 18.3 | 32.0 |
| Lautoka | N/A | 9.8; 25.6 |
| Lenakel | 6.1; 8.5 | 5.6; 12.8 |
| PV | N/A | 25.6 |
| Ouinne | 8.5; 25.6 | 9.1; 32.0 |
| Thio | N/A | 8.5; 14.2 |
| Hienghene | N/A | 7.5; 18.3 |
| KJN | 8.5; 14.2 | 14.2 |
| RIFR | 4.6; 9.1 | N/A |

---

## Author Comment (AC3)

**Response to RC2**

7th January 2022

Dear Reviewer,

We sincerely thank you for the constructive comments that greatly helped us to improve the manuscript. Here we present our point-by-point responses and revision to the comments.

Sincerely,

Yuchen Wang
Postdoctoral Researcher,
Japan Agency for Marine-Earth Science and Technology,
Japan
Email: ywang@jamstec.go.jp

**General Review Comments**

*This paper describes the source spectrum of the tsunamis generated by two earthquakes (Mw 7.4 and Mw 8.1) that occurred in the Kermadec subduction zone on 4 March 2021 using tsunami data recorded at tide gauges. The study produced the dominant wave period range for each event and the spectral ratio for the larger earthquake (second event) by utilizing the data from the smaller event (first event) as Green's Functions.*

**Comment 1**

*High-quality water level data at DART stations are available for this event. The DART stations are in the deep ocean which means the records are not affected by the coastal geomorphology, unlike tide gauges. It is not clear why the study is only using tide gauge but not DART data. I suggest the inclusion of spectral analysis of DART data in this study.*

**Response and Revision**

We acknowledge that DART data provides important information. In general, the 1 min (or 15 s) data of DARTs are available for short times covering only a few tsunami waves and the rest of DART data has a large sampling interval of 15 min. Therefore, the length of DART data with 1 min (or 15 s) sampling is short, which cannot meet the requirement of spectral analyses. The spectral analyses of low-sampling data will miss high-frequency components. Therefore, we decided not to include DART data. In response to this comment, we added the reason.
In Line 73, we added:

'We did not use the data from Deep-ocean Assessment and Reporting of Tsunamis (DARTs) tsunameters because the duration of high sampling mode (1 min or 15 s) is not long enough for spectral analyses and as a result, the spectra would be unreliable. We note that the DART data of these events were published by Romano et al. (2021).'

**Comment 2**

*The paper provides the dominant wave period ranges for the first and second events. But what the meaning of those numbers is not explained. The dominant period ranged from 8 to 28 minutes for the Mw8.1 earthquake (the range is 7-28 min in lines 175-180, which one is the correct one). Is this a normal range for this kind of earthquake?*

**Response and Revision**

Thank you for this comment. The energy ranges are slightly different when being calculated by different analyses (8–32 min for Fourier analysis, 8–30 min for EGF method, and 7–28 min for second/background noise), but they are generally consistent. In response to this comment, we added the following statements in the revised manuscript.
In Line 143, we added:

"Tsunami spectra can help identify the source size, potential asperities and other information about earthquake source processes. The dominant wave period ranges for tsunami events are related to the size of the source, which we explain in Section 5. Assuming the same water depths, tsunamis generated by earthquakes with larger source sizes normally have longer dominant wave periods. For example, the tsunami generated by the Mw 9.1 2004 Sumatra earthquake in the near-field region indicated dominance of long waves with periods of 30–60 min (Rabinovich et al., 2006)."

**Comment 3**

*It is not clear how the period range was determined. Was it from the Fourier or the wavelet analysis? Lines 130-135 describe that the range was determined from the peak spectral power. But the peak for the first event at some of the stations like Owenga is longer than the upper limit of the 5-17 min range. Please indicate the peak at each station for each event with an inverted triangle in Figure 3.*

**Response and Revision**

We chose the period ranges that are present and dominate in more than half of the stations. We acknowledge that the peak period range may not be exactly the same among all stations. In response to this comment, we made following revisions:
(1) We revised Figure 3, indicating the peak at each station for each event. We also listed the values in Table 1.

(2) In the caption of Figure 3, we added:

'Green dots show the spectral peaks listed in Table 1.'

(3) In Line 139, we changed:

'…the dominant periods range for the second tsunami is approximately  8–32 min.'

(4) In Line 141, we added:

'Regarding tsunami dominant period (or peak periods), here we chose the period range that more than half of the stations present as the dominant range.'

[Figure]

**Figure 3: Fourier analyses for tsunamis generated by two successive earthquakes (M$_w$ 7.4 and M$_w$ 8.1) in Kermadec Islands. Pink and red curves represent the spectra of the first tsunami and the second tsunami, respectively. Green dots show the spectral peaks listed in Table 1. The 95% confidence bounds of two tsunami spectra are indicated by dashed curves. The background spectra (black curves) are also plotted for comparison.**

**Table 1: Peak periods at each tide gauge for two tsunami events. The values were calculated by Fourier analyses. Station name abbreviations are: North Cape (NC), Great Barrier Island (GBI), East Cape (EC), Suva Viti Levu (SVL), Kingston Jetty Norfolk (KJN), Port Vila (PV), and Raoul Island Fishing Rock (RIFR).**

| Tide gauge | Peak period(s) for the first tsunami (min) | Peak period(s) for the second tsunami (min) |
|---|---|---|
| NC | 9.1 | 9.8; 21.3 |
| GBI | 6.5; 10.7 | 6.4; 10.7; 32.0 |
| PT | N/A | 9.8 |
| EC | 6.1; 9.8; 16.0 | 8.5; 18.3 |
| Owenga | N/A | 14.2 |
| Nukualofa | N/A | 7.1; 21.3 |
| SVL | 8.0; 18.3 | 32.0 |
| Lautoka | N/A | 9.8; 25.6 |
| Lenakel | 6.1; 8.5 | 5.6; 12.8 |
| PV | N/A | 25.6 |
| Ouinne | 8.5; 25.6 | 9.1; 32.0 |
| Thio | N/A | 8.5; 14.2 |
| Hienghene | N/A | 7.5; 18.3 |
| KJN | 8.5; 14.2 | 14.2 |
| RIFR | 4.6; 9.1 | N/A |

**Comment 4**

*Similarly, the period range determination using the wavelets is also not so clear. The period range for the second event detected at Ouinne was 20-30 min. But the paper finally concluded the period range of 8-28 min for the second event. Was the period at Ouinne simply ignored? Moreover, if we chose the peaks in the Ouinne wavelet using the contours, we would get a range of 10-30 min instead of 20-30 min. Please provide a table with the range for each station.*

**Response and Revision**

Thank you for this observation. We agree that adding a table of peak periods will help and therefore we added a table. Similar to the response to Comment 3, we chose the period range that more than half of the stations present as the dominant range, instead of a single station. We acknowledge that the peak period range may not be exactly same among all stations. In addition, we also note that wavelets and Fourier analyses give spectral results with varying degrees of accuracies. In response to this comment, we made the following revisions:

(1) We added a table with the range for each station (Table 1).

(2) In Line 143, we added:

'In Table 1, we listed the peak periods at each tide gauge for the two tsunamis.'

(3) In Line 175, we added:

'In addition, we also note that wavelets and Fourier analyses give spectral results with varying degrees of accuracies, because wavelet analysis also incorporates the time evolution and thus its spectra are not usually as detailed as those obtained by Fourier analyses.'

**Comment 5**

*Figure 4: Provide purple and red boxes for the other tide stations.*

**Response and Revision**

We acknowledge that this item would add to the clarity of the paper. Therefore, we added purple and red boxes to stations NC, GBI, Lenakel, Ouinne, and KJN.

[Figure]

**Figure 4**: **Wavelet (frequency-time) analyses for tsunamis generated by two successive earthquakes (M$_w$ 7.4 and M$_w$ 8.1) in Kermadec Islands. The colormap shows levels of spectral energy at different times and periods. For guidance, we marked the dominant periods of two tsunamis by pink (M$_w$ 7.4) and red (M$_w$ 8.1) rectangles** . **The pink and red vertical lines show the origin times of the M$_w$ 7.4 and M$_w$ 8.1 earthquakes, respectively. The dispersion curves are plotted by black dashed lines. On the horizontal axis, plus one (+1) indicates one day passed.**

**Comment 6**

*Lines 185-190: the paper argues that the spectral analysis validates the USGS source model. But the calculation of the model source periods used the total fault length and width instead of the dimension of the main slip region, which is about 120 km long and 120 km wide. Outside this main slip region, the slip amounts are almost zero, so the total fault dimension should not be used in the calculation.*

**Response and Revision**

We agree. Here the *L* in Equation (1) is the size of tsunami source. The size is related to the surface displacement, which may not be equal to the size of fault slip region. In the revised manuscript, we consider only the non-zero displacement region as reviewer suggest. Our investigation of the USGS source model shows that the non-zero displacement region is approximately 210 km x 170 km (https://earthquake.usgs.gov/earthquakes/eventpage/us7000dflf/finite-fault).
In Line 213, we changed:

'However, the non-zero displacement region is approximately 210 km $\times$ 170 km (https://earthquake.usgs.gov/earthquakes/eventpage/us7000dflf/finite-fault). The average water depth in the source area is ~ 5,000 m. Hence, the first three source periods of the short axis (width) using the analytical equation (Eq. 1) are  calculated as 25.6 min, 12.8 min, and 8.5 min.  These values are consistent with the results of spectral analyses of the observed waves based on the EGF and tsunami-to-background spectral ratio methods (Figures 5c; 5d) showing peak periods at 25.6 min, 16.0 min and 9.8 min (8.5 min).'

**Comment 7**

*Provide the spectral ratios from the simulated waveforms and then compare them with the observed ones.*

**Response and Revision**

We agree that it is useful to compare simulated tsunami spectra with observed ones. However, we lack detailed bathymetry around the tide gauges and thus it is difficult to accurately simulate the waveforms at each station. Hence, we are not sure whether adding simulated spectra would help and therefore in this study we only rely on observed tsunami waveforms for spectral analyses.

**Comment 8**

*Error bars are required for Figure 5c and 5d.*

**Response and Revision**

In response to this comment, we added error bars to Figures 5c and 5d.

[Figure]

**Figure 5: (a, b) Maximum simulated amplitudes for two tsunamis during the entire simulation time. The source models used for numerical simulation are from the USGS. (c) Spectral ratio of two tsunamis by dividing the spectral energy of the second tsunami to that of the first tsunami (EGF method). Blue curve is the normalized average of tsunami spectral energy at different tide gauges. (d) Spectral ratio of the second tsunami spectrum to the background signal spectrum. Green curve is the normalized average of different tide gauges. (e) Spectral ratio of the first tsunami spectrum to the background signal spectrum. Green curve is the normalized average of different tide gauges.**

---

## Author Comment (AC4)

**Response to RC3**

31st January 2022

Dear Reviewer,

We sincerely thank you for the constructive comments that greatly helped us to improve the manuscript. Here we present our point-by-point responses and revision to the comments.

Sincerely,

Yuchen Wang
Postdoctoral Researcher,
Japan Agency for Marine-Earth Science and Technology,
Japan
Email: ywang@jamstec.go.jp

**General Review Comments**

*In the manuscript entitled 'Characteristics of two tsunamis generated by successive Mw 7.4 and Mw 8.1 earthquakes in Kermadec Islands on March 4, 2021', the authors have addressed a not trivial case capturing tsunami characteristics generated by two successive earthquakes. This study provides insights of the source spectrum based on the empirical Green's function (EGF) and tsunami/background ratio methods. The spectral analysis allows distinguishing the dominant wave periods and ranges. The paper is well written but the discussion and conclusions should be improved. I suggest further revision to the following comments.*

**Comment 1**

*While the selection of a second-order high-pass filter has been tested (e.g. Heiderzadeh and Satake, 2013; Heiderzadeh et al., 2015), the justification of the window or frequency corner is not clearly stated. This selection may affect results.*

**Response and Revision**

Thank you. As the maximum tsunami period in our study is approximately 30 min, it is reasonable to apply a filter with a corner period of four times longer than that value.

**Comment 2**

*Considering that this is a peculiar case where two tsunamis are generated from close sources, I've found that the Section 5 and the Conclusions are weak. These sections*

*could be enhanced by exploiting more the results, and providing a thorough discussion that is lacking. Also, consider to add limitations of this study (e.g. not using the DART records).*

**Response and Revision**

Thank you very much. In response to this comment, we added a new paragraph at the end of Section 5.

In Line 246, we added:

"As limitations of this study, we could mention a few items: We are not using DART data due to the short duration of high-sampling records. In general, DART records are valuable type of sea level data in terms of tsunami source studies because they are less affected by local and regional bathymetry. In addition, the number of sea level records that we used for analyses of these tsunamis is not very large due to the limited number of available stations."

**Comment 3**

*The authors are encouraged to provide further interpretation of the results higher frequencies, decaying processes and source characteristics, specially in light of one of the main observations of this study described in Line 195.*

**Response and Revision**

We acknowledge. There are short-period (high-frequency) waves of approximately 5 min, which arrives at later stages due to dispersion. In Section 4, we added the following descriptions:

In Line 175:

"We note that the dispersive effects of tsunamis from the second event are evident on the wavelet plots as tsunami dominant period for the few initial waves is around ~20 min, whereas it linearly shifts towards ~10 min for the later waves, giving us the opportunity to plot the inverse dispersion lines (black dashed lines in Figure 4). We plotted the dispersion curve on these diagrams. We also observe short-period waves with period of 5–8 min at some sea-level stations (Table 1; Figures 3–4), which we attribute to various local bathymetric effects. In addition, we also note that wavelets and Fourier analyses give spectral results with varying degrees of accuracies, because wavelet analysis also incorporates the time evolution and thus its spectra are not usually as detailed as those obtained by Fourier analyses."

**Comment 4**

*The overall structure of the paper is fine, but some elements of the Methods appear in the Section 5, where instead, discussion is expected. Also, the results of the subsection 2.3 'Earthquake Slip Models and Tsunami Numerical Simulation', where simulations*

*based on the USGS source models need further contextualization/discussion, for example, in Figure 5.*

**Response and Revision**

(1) We agree. In response to this comment, we moved some content from Section 5 and created a new Section 2.4.

In Line 113, we added:

"2.4 Calculating Tsunami Source Period

In this study, we calculated tsunami source period from finite fault models to compare with the results of spectral analyses. Theoretically, the tsunami source period is related to earthquake rupture length and water depth (Rabinovich, 1997; 2010; Heidarzadeh and Satake, 2013; Wang et al., 2021). It can be estimated as:

$$T_n = \frac{2L}{n\sqrt{gh}} \quad n = 1,2,3, \dots \quad , \tag{1}$$

where $L$ is the typical size of tsunami source area (length or width), $g$ is the gravity acceleration, and $h$ is the average water depth in source area."

(2) We added more explanations on our simulations based on the USGS model.

In Line 204, we added:

"We simulated the propagation of two tsunamis using *JAGURS* code and plotted their maximum amplitude in our region of interest to investigate their propagation path (Figures 5a and 5b). , and the The tsunami amplitude in the NW-SE direction is larger than that in the NE-SW direction because it is parallel to the short axis of the fault. The propagation paths of two successive tsunamis are similar (Figures 5a and 5b)."

**Comment 5**

*Lines 83-84, articles needed.*

**Response and Revision**

In response to this comment, we added some reference articles.
In Line 88, we added:

"We ensured that there were no storms or other atmospheric events at the time period of the background signals, so the background spectra could exclusively reflect the frequency response of local topography (Cortés et al., 2017; Aránguiz et al., 2019). Tidal components were removed by applying a high-pass filter in a similar way to preparation of the tsunami records (Heidarzadeh and Satake, 2013)."

**Comment 6**

*Line 127, needs rephrasing: Similar patterns were also be observed at Ouinne…*

**Response and Revision**

In response to this comment, we added more explanations.
In Line 140, we added:

"Similar patterns were also be observed at Ouinne: Short-period components existed in the few hours after the second tsunami's arrival, but the waveforms after 01:00 (+1) were dominated by long-period components (Figure 2)."

**Comment 7**

*Line 133, modify 'at most stations. At most stations …'*

**Response and Revision**

In response to this comment, we modified the language.
In Line 148, we changed:

" The peak periods of the first tsunami are mostly distributed in the range of 5–17 min…"

**Comment 8**

*The figures of this manuscript have been greatly improved in the answers to previous comments, but I suggest to modify the green point that shows the spectral peaks in Figure 3. Instead, consider using a stronger color.*

**Response and Revision**

In response to this comment, we changed the color of points showing the spectral peaks.